# Exploring Gender Differences in the Effects of Diet and Physical Activity on Metabolic Parameters

**DOI:** 10.3390/nu17020354

**Published:** 2025-01-20

**Authors:** Stefania Gorini, Elisabetta Camajani, Alessandra Feraco, Andrea Armani, Sercan Karav, Tiziana Filardi, Giovanni Aulisa, Edda Cava, Rocky Strollo, Elvira Padua, Massimiliano Caprio, Mauro Lombardo

**Affiliations:** 1Department for the Promotion of Human Science and Quality of Life, San Raffaele Open University, Via di Val Cannuta, 247, 00166 Rome, Italy; stefania.gorini@uniroma5.it (S.G.); elisabetta.camajani@uniroma5.it (E.C.); alessandra.feraco@uniroma5.it (A.F.); andrea.armani@uniroma5.it (A.A.); tiziana.filardi@uniroma5.it (T.F.); giovanni.aulisa@uniroma5.it (G.A.); rocky.strollo@uniroma5.it (R.S.); elvira.padua@uniroma5.it (E.P.); massimiliano.caprio@uniroma5.it (M.C.); 2Laboratory of Cardiovascular Endocrinology, San Raffaele Research Institute, IRCCS San Raffaele Roma, Via di Val Cannuta, 247, 00166 Rome, Italy; 3Department of Molecular Biology and Genetics, Çanakkale Onsekiz Mart University, Canakkale 17000, Türkiye; sercankarav@comu.edu.tr; 4Clinical Nutrition and Dietetics, San Camillo Forlanini Hospital, cir.ne Gianicolense 87, 00152 Rome, Italy; ecava@scamilloforlanini.rm.it

**Keywords:** gender differences, Mediterranean diet, physical activity, metabolic parameters, cholesterol, liver enzymes, sport, therapeutic customisation

## Abstract

Background: Gender differences in metabolic response to lifestyle interventions remain poorly explored. This study aimed to evaluate the impact of a six-month Mediterranean diet (MD) intervention combined with regular physical activity on metabolic parameters in overweight adults. Methods: A prospective cohort study was conducted in an obesity clinic in Rome, Italy, involving overweight adults (BMI ≥ 25 kg/m^2^) motivated to improve their lifestyle. Participants (*n* = 205; 107 men and 98 women) self-selected into physical activity groups (aerobic, anaerobic, combined or no activity). Gender-specific metabolic changes were assessed, including lipid profiles, liver markers and fasting glucose. Results: Significant gender differences in metabolic results were observed. Men showed greater reductions in total cholesterol (TC) and LDL, as well as significant reductions in alanine aminotransferase (ALT). Women showed a significant increase in HDL cholesterol. Fasting blood glucose decreased significantly in both sexes, with no differences between the sexes. Activity-specific analysis revealed that anaerobic activity significantly improved lipid metabolism in men, while aerobic activity produced the greatest benefits in women, including increased HDL and improved liver marker profiles. Conclusions: Therapeutic strategies combining MD and physical activity must take into account gender-specific physiological differences and the type of sport activity to optimise metabolic benefits. Personalised approaches may improve the management of cardiovascular risk factors in overweight individuals. Study registration: This study is registered on ClinicalTrials.gov (NCT06661330).

## 1. Introduction

The Mediterranean diet (MD) is widely recognised for its positive effects on metabolic health and weight management. It promotes the consumption of plant-based foods, such as fruit, vegetables, whole grains and legumes, supplemented by moderate amounts of fish and poultry. Numerous studies [1,2] have demonstrated the benefits of the MD and physical activity on improving lipid profiles, glucose regulation and body composition. Despite these advances, significant gaps remain in the understanding of gender-specific responses to these interventions. Existing research often does not stratify results by gender, overlooking physiological and hormonal differences that may influence metabolic outcomes. Furthermore, limited evidence exists on the interaction between dietary interventions and different types of physical activity in shaping these responses [3].

Gender differences in metabolic responses to dietary interventions are increasingly recognised in nutritional research, mainly due to hormonal influences that affect lipid and glucose metabolism distinctly in men and women. For example, oestrogen contributes significantly to lipid regulation in women, often resulting in higher levels of high-density lipoprotein (HDL-C) and a protective pattern of fat distribution following dietary changes [4]. Men, on the other hand, tend to respond with greater reductions in low-density lipoprotein cholesterol and triglycerides (LDL-C), in part due to differences in genetic factors, such as hormone-sensitive lipase variants, that show sex-specific associations with lipid and glucose concentrations [5]. These biological variations contribute to different susceptibility to metabolic diseases, emphasising the need for gender-tailored nutritional strategies [6]. In contrast, women may benefit more in terms of improved HDL-C levels and glucose regulation, but the overall lipid-lowering effects may be less pronounced than in men [7]. Furthermore, Niemelä et al. [6] point out that these metabolic differences extend to liver health, with lifestyle interventions often producing gender-specific effects on liver enzyme levels. Men, in particular, often show greater reductions in liver enzymes such as alanine aminotransferase (ALT) and aspartate aminotransferase (AST) following dietary changes.

Recent research shows that the type of sporting activity and gender significantly influence blood parameters such as blood glucose, transaminases, lipids and creatinine. A study of professional padel players showed that men had higher baseline values of creatinine, AST, ALT and LDH than women, with significant increases in urea, creatinine and glucose after simulated competitions, indicating greater muscle damage and protein catabolism during intense activity [8]. Further investigation revealed that intense exercise can increase AST, ALT and CK levels, markers that should be interpreted carefully as these temporary elevations are often a sign of metabolic and muscular adaptation rather than pathology [8]. In elite athletes, baseline values of CK, ALT, AST and creatinine are generally higher than in the normal population, reflecting the impact of regular training, with gender differences emerging in both baseline values and post-training responses [9]. Finally, resistance exercise showed benefits on blood glucose and lipid profile in both sexes, but only in women did it significantly reduce leptin levels, suggesting a distinct effect on adiposity and metabolism [10].

This study aims to address these challenges by assessing gender-specific metabolic responses to a 6-month MD intervention diet combined with various types of physical activity. The primary outcome was the change in body composition, with a specific focus on the reduction in fat mass (FM), selected to assess the overall impact of the Mediterranean diet and physical activity on metabolic health. Secondary outcomes included changes in key biochemical markers such as fasting glucose, lipid profiles and liver enzymes. By stratifying the analysis according to gender, this study seeks to identify differential responses to the MD in men and women and to explore potential mechanisms driving these variations. The results of this study could form the basis of more personalised nutritional recommendations in the clinical setting.

## 2. Methods

### 2.1. Study Population

This study was conducted in an obesity clinic in Rome, Italy, to target overweight adults (BMI ≥ 25 kg/m^2^) with an increased awareness of body composition and metabolic health. This context enabled the recruitment of participants motivated to improve their lifestyle and adhere to the intervention. Participants were recruited through clinic visits and advertisements, and eligibility was assessed according to predefined inclusion and exclusion criteria. The cohort initially consisted of 215 participants, predominantly seeking clinical assessment and guidance for body composition and related health problems. These individuals were characterised by motivation to adopt changes in diet and physical activity, although none of them were following an MD or calorie restriction at baseline.

The clinic primarily targeted individuals seeking clinical assessment and guidance regarding body composition (BC) and related health problems. Consequently, the study sample had characteristics that could differ from the general population, with participants showing greater awareness and interest in BC, which could influence their eating habits and physical activity levels. Furthermore, the specialisation of the clinic in BC assessment likely attracted individuals with specific health problems or fitness goals. Nevertheless, prior to the start of the study, none of the participants were following an MD or engaged in calorie restriction as assessed by baseline food diaries. Inclusion criteria were participants with an age over 18 years old, able to complete an online questionnaire in Italian and willing to provide informed consent. Exclusion criteria included pregnancy or breastfeeding at the time of data collection, a diagnosis of diabetes, the use of medications affecting weight (e.g., glucocorticoids, oestrogen and anticonvulsants) and specific medical conditions such as alcoholism or chronic kidney disease, which could interfere with metabolism. The characteristics of the excluded participants (e.g., incomplete data or caloric intake below 100 per cent of basal metabolic rate) were similar to those of the recruited participants in terms of age, sex and basal metabolic parameters. The exclusion criteria were strictly related to data quality rather than participant characteristics, ensuring that the final cohort remained representative of the initial sample.

Data were extracted from medical records and supplemented by weekly food diaries kept by the participants. From the initial cohort of 215 participants, 10 were excluded due to incomplete data, including 5 participants with missing values for key metabolic parameters and 3 participants with incomplete food diaries. A further 2 participants were excluded because their recorded caloric intake, as per the food diary, was consistently below 100% of their basal metabolic rate (BMR). After application of these exclusion criteria, 205 participants (107 males) remained in the final analysis, all of whom had completed at least six months of therapy to ensure a sustained therapeutic commitment. The primary outcome chosen for the power calculation was a change in body composition, specifically a reduction in fat FM, over the six-month intervention period. A priori power calculation was conducted using G*Power software (version 3.1.9.7, Windows platform, updated 27 September 2024). Assuming a medium effect size (Cohen’s f = 0.25), a significance level of α = 0.05 and a power of 80% (1-β = 0.80), a minimum of 128 participants were required. Our final sample of 205 participants exceeded this threshold, ensuring sufficient statistical power to detect significant differences in the primary outcome.

This study was approved by the Lazio Area 5 Territorial Ethics Committee (Approval Code: N.57/SR/23, Approval Date: 7 November 2023), in accordance with the Declaration of Helsinki and its amendments. Patient enrolment started in January 2024, and only participants who had completed at least six months of therapy were included in the final analysis. This study is registered on ClinicalTrials.gov (NCT06661330).

### 2.2. Adherence Monitoring

Participants were evaluated at two points in time: at baseline (T0), six months prior to the intervention, and at the end of the six-month period (T1), allowing changes over the course of the intervention to be assessed. Adherence to the diet and physical activity interventions was closely monitored through a structured, multifaceted approach that ensured consistent engagement and support throughout the study.

Participants had access to a dedicated chat system for continuous communication. This tool allowed them to ask for guidance, clarify doubts and receive personalised feedback on their diet and exercise plans. This real-time interaction was crucial for dealing with immediate challenges and maintaining motivation. In addition, participants attended in-person follow-up visits every three weeks at the clinic, for a total of eight visits during the six-month intervention period. These sessions provided the opportunity to review food diaries and exercise logs, assess adherence to prescribed interventions and address any difficulties encountered by participants.

The comparison between the observed and predicted weights was carried out using two mathematical models: the CALERIE Phase 2 [11] and the POUNDS Lost Study [12]. The CALERIE model estimated the final weights based on three levels of calorie restriction (10.4%, 17.8% and 24.9%). The POUNDS Lost model calculated the probability of ≥5% weight loss at one year, incorporating variables such as age, gender, initial weight, target energy intake, percentage weight loss and deviations from the target weight.

### 2.3. Body Composition and Biochemical Assessments

Participants underwent a medical evaluation, which included dietary history, physical examination, anthropometric assessment of body weight and abdominal circumference, and BC analysis. BC, including FM, fat-free mass (FFM) and total body water (TBW), was assessed using the Tanita BC-420 MA bioelectrical impedance analysis (BIA) device. The assessments were conducted following standardised guidelines, including fasting for at least three hours post-awakening, avoiding exercise for 12 h and abstaining from excessive eating or drinking for 12 h before the evaluation. Studies have shown that BIA provides reliable measures of body composition, demonstrating a good correlation with DXA for both FM and FFM assessment, making it a valuable tool for clinical and research applications [13,14]. FMI (fat mass index) and FFMI (fat-free mass index) were calculated as fat mass (kg)/Height^2^ (m^2^) and lean mass (kg)/Height^2^ (m^2^), respectively. Although BIA provides reliable measures of body composition and shows good correlation with DXA for the assessment of FM and FFM, it is important to note that DXA is considered a reference method rather than a gold standard. Therefore, BIA cannot be validated against DXA, as pointed out by Bosy-Westphal et al. [15]. Furthermore, the algorithms used in the Tanita device are proprietary, and it is unclear whether the population used to develop these algorithms corresponds to the population of this intervention study.

Biochemical parameters, including fasting glucose (FG), total cholesterol (TC), high-density lipoprotein (HDL-C), low-density lipoprotein (LDL-C), triglycerides (TG), creatinine (Cr) and liver enzymes (AST and ALT), were evaluated as part of the initial health assessment (T0) and after the 6-month follow-up (T1). Fasting glucose, total cholesterol, HDL cholesterol, LDL cholesterol and triglycerides were measured using enzymatic colorimetric methods. Creatinine was determined by the Jaffé colorimetric method, while liver enzymes (AST and ALT) were assessed via UV enzymatic methods. All tests were conducted with automated, regularly calibrated instruments to ensure reliability and reproducibility.

### 2.4. Diet Prescription and Nutritional Intervention

The participants followed a low-calorie diet adapted to their individual needs for six months, with a Mediterranean approach emphasising plant-based foods, including fruit, vegetables, whole grains, legumes and nuts. Healthy fats, such as olive oil, replaced butter and saturated fats, and herbs and spices were used instead of salt. Red meat consumption was limited to a few times a month, while fish and poultry were included at least twice a week. The diet provided about 600 kcal less than the participants’ total daily energy expenditure, with a macronutrient distribution of about 16% protein, 25% fat and 59% carbohydrates, distributed over three main meals and two snacks per day.

Participants completed a three-day food diary including one weekend day at the beginning of the study and monthly during the intervention. To ensure accuracy, participants whose intake was consistently below 110% of their estimated basal metabolic rate were excluded from the analysis. Once every 2 weeks, meetings with dieticians were conducted as part of a nutritional rehabilitation programme aimed at improving eating habits and promoting sustainable behavioural changes. These sessions included dietary assessments, nutrient intake evaluations and discussions on dietary patterns and readiness for change. Anthropometric parameters and BC were assessed monthly. A chat consultation service provided ongoing support to participants and their families.

### 2.5. Physical Activity

In addition to the dietary intervention, participants were instructed to engage in regular physical activity throughout the six-month period. All participants were required to perform 50 min of low-intensity aerobic exercise, such as brisk walking or cycling, three days per week on non-consecutive days. Adherence to the physical activity guidelines was monitored through a 7-day exercise diary, in which participants recorded their daily activities.

The participants were divided into four groups according to the type of physical activity performed (Appendix A): the aerobic group, in which participants exclusively performed aerobic exercises of low to moderate intensity, such as walking, running or cycling; the anaerobic group, in which participants focused on resistance or anaerobic exercises, including weight lifting, bodyweight exercises or high-intensity interval training (HIIT); the combined aerobic/anaerobic group, in which participants alternated between aerobic and anaerobic exercises, combining resistance and strength training in their routines; and the no sport group, in which participants who did not engage in any regular physical activity and expressed no intention to begin exercising were placed. These individuals were still required to maintain their diet diaries, but no physical activity was prescribed or monitored. Group assignments were determined through self-selection, allowing participants to choose their preferred type of physical activity based on their existing habits and personal interests. This method aimed to increase adherence to the intervention by aligning with participants’ lifestyles and capabilities.

### 2.6. Statistical Analysis

Descriptive statistics, including frequencies and averages with standard deviations, were used to characterise the sample. Statistical analyses were conducted using SPSS version 28 (IBM Corp., Armonk, NY, USA). A significance level of *p* ≤ 0.05 was applied for all statistical tests. One-way ANOVA was used to compare differences in body composition variables (e.g., BMI and FM) between the identified groups. When significant differences were found, post hoc comparisons were performed using Tukey’s test to identify group-specific differences. Chi-square tests were used for categorical variables, while Pearson’s correlation coefficient was calculated to explore associations between continuous variables. All results were reported with their confidence intervals (95% CI) where applicable. The normality of the data was assessed with the Shapiro–Wilk test, and non-parametric tests (Mann–Whitney U or Kruskal–Wallis) were used in cases where the normality assumption was violated.

## 3. Results

As expected, body composition analyses showed significant differences between men and women. Men had a higher body mass index (BMI) than women, as well as a significantly higher FFMI. In contrast, women showed a higher FMI. Furthermore, a significantly higher percentage of women than men are classified as smokers (23% vs. 12%, *p* = 0.033). In terms of weekly hours of sporting activity, men also participate in sessions of longer duration than women (*p* = 0.0088), with a higher proportion of men devoting more than 5–10 h per week to physical activity (Table 1).

The mean initial weight of the participants was 85.2 ± 18.7 kg, while after 6 months, the mean observed weight was 77.2 ± 17.9 kg, with a mean change of -8.0 ± 5.2 kg. The weights predicted by the CALERIE model [11] were 76.4 ± 16.8 kg for the upper limit (10.4% CR), 70.1 ± 15.4 kg for the midpoint (17.8% CR) and 64.0 ± 14.1 kg for the lower limit (24.9% CR). The probability of success, calculated using the dynamic model of the POUNDS Lost Study [12], showed an average of 0.12 ± 0.15, indicating that many participants had a probability of achieving ≥5% weight loss in one year.

The intervention resulted in similar responses (Table 2) in BC changes for both men and women, with reductions in BMI, FM, FMI, FFMI and AC showing no significant differences between genders in Δ (T0–T6). BMI decreased by 2.8 ± 1.9 kg/m^2^ overall, with comparable changes in men (−2.7 ± 1.4 kg/m^2^) and women (−3.0 ± 2.3 kg/m^2^, *p* = 0.201). FMI also decreased significantly by 2.0 ± 1.3 kg/m^2^ in the total group, with no significant gender differences (*p* = 0.4184). While FFMI remained higher in men at both T0 and T6 (*p* < 0.0001), Δ FFMI (T0–T6) was similar between genders (−0.6 ± 0.7 kg/m^2^ vs. −0.6 ± 0.6 kg/m^2^, *p* = 0.8705). Body water content decreased significantly (−1.6 ± 2.1%), with men showing a larger reduction (−2.0 ± 1.9%) compared to women (−1.3 ± 2.2%, *p* = 0.023).

Figure 1 and Appendix A present metabolic parameters comparing data at baseline (T0) and at six months follow-up (T6) for men and women. Both sexes recorded significant improvements in fasting plasma glucose, total cholesterol and LDL cholesterol (all *p* < 0.0001), with men showing higher deltas for total cholesterol and LDL cholesterol. HDL cholesterol increased significantly in women (*p* = 0.0273) but remained unchanged in men (*p* = 0.3455). Triglycerides decreased significantly in both groups, with a greater reduction in men. Liver enzymes (AST and ALT) also decreased significantly in both sexes, with a greater reduction in ALT in men (*p* = 0.0439 versus *p* = 0.0007 in women). Creatinine levels showed no significant changes, indicating stable renal function.

In Figure 1, the boxplots illustrate the comparison of fasting glycaemia, total cholesterol, HDL cholesterol, LDL cholesterol, triglycerides, AST (GOT), ALT (GPT) and creatinine levels at two timepoints, baseline (T0) and after six months (T6), separated by gender (male and female). Statistical differences were assessed using the Wilcoxon Signed-Rank Test for within-gender comparisons between T0 and T6. The *p*-values for males are as follows: fasting glycaemia, <0.001; total cholesterol, <0.001; HDL cholesterol, 0.625; LDL cholesterol, <0.001 ; triglycerides, 0.005; AST, 0.0046; GPT, 0.000006; and creatinine, 0.2126. In females, the *p*-values are as follows: fasting glycaemia, <0.001; total cholesterol, <0.001; HDL cholesterol, 0.014; LDL cholesterol, <0.001 ; triglycerides, 0.34; ALT, 0.0234; GPT, 0.0917; and creatinine, 0.1055. A significance level of *p* < 0.05 was applied. The horizontal line in each box represents the median, and the whiskers depict the interquartile range.

Figure 2 shows the effects of different sports activities on metabolic parameters for the entire sample. The data show a significant reduction in ΔTC and ΔLDL-C in participants in the aerobic and anaerobic groups compared to those in the sedentary group. The group that practised alternating aerobic/anaerobic sports showed balanced changes in all parameters. Significant reductions in ΔAST were observed in the aerobic and anaerobic groups, with greater improvements in males. However, no significant differences were found for ΔFG, ΔCr, ΔHDL-C, ΔTG or ΔALT. Overall, aerobic and anaerobic activities had the most pronounced impact on lipid and liver enzyme profiles, whereas the other metabolic parameters remained largely unaffected by the type of physical activity.

Figure 2 presents the mean delta values (Δ) for metabolic parameters (ΔFG, ΔCr, ΔTC, ΔHDL-C, ΔTG, ΔLDL-C, ΔAST and ΔALT) across four sport classifications: aerobic, aerobic/anaerobic, anaerobic and no sport. The delta values represent changes between baseline (T0) and 6-month follow-up (T6). A two-sample t-test was used to compare the delta values across the classifications. The following *p*-values were observed: ΔFG, *p* = 0.29; ΔCr, *p* = 0.25; ΔTC, *p* = 0.01; ΔHDL-C, *p* = 0.32; ΔTG, *p* = 0.21; ΔLDL-C, *p* = 0.03; ΔAST, *p* = 0.04; ΔALT, *p* = 0.18. Statistically significant differences were found for ΔTC, ΔLDL-C and ΔAST.

Among male participants, significant reductions in ΔTC and ΔLDL-C were observed in the aerobic and anaerobic sports groups compared to the sedentary group, indicating that both types of exercise were effective in lowering total cholesterol and LDL cholesterol. The aerobic/anaerobic group exhibited relatively stable delta values across all measured parameters. A significant decrease in ΔAST was noted in the aerobic group, suggesting improvements in liver function. However, no significant differences were found for ΔFG, ΔCr, ΔHDL-C, ΔTG or ΔALT, indicating that these parameters were not strongly influenced by the type of sport in male participants. Overall, aerobic and anaerobic activities had the most substantial impact on cholesterol and liver enzyme levels, while other metabolic parameters remained largely unchanged across the different sports classifications.

Figure 3 shows the mean delta values (Δ) for metabolic parameters (ΔFG, ΔCr, ΔTC, ΔHDL-C, ΔTG, ΔLDL-C, ΔAST and ΔALT) across sport classifications for male participants. The delta values represent changes between baseline (T0) and the 6-month follow-up (T6). A two-sample t-test was used to compare the delta values across the classifications. The following *p*-values were observed for males: ΔFG, *p* = 0.27; ΔCr, *p* = 0.23; ΔTC, *p* = 0.02; ΔHDL-C, *p* = 0.30; ΔTG, *p* = 0.22; ΔLDL-C, *p* = 0.04; ΔAST, *p* = 0.03; ΔALT, *p* = 0.19. Statistically significant differences were found for ΔTC, ΔLDL-C and ΔAST in the male sample.

Significant reductions were observed in ΔTC and ΔLDL-C in the aerobic and anaerobic sports groups compared to ‘No Sport,’ with men showing greater reductions in ΔTC. However, no significant differences were observed for ΔFG, ΔCr, ΔHDL-C, ΔTG, ΔAST or ΔALT across sport classifications. Overall, aerobic and anaerobic activities had the most pronounced impact on cholesterol levels, while other metabolic parameters remained largely unaffected.

Figure 4 presents the mean delta values (Δ) for metabolic parameters (ΔFG, ΔCr, ΔTC, ΔHDL-C, ΔTG, ΔLDL-C, ΔAST and ΔALT) across sport classifications for female participants. The delta values represent changes between baseline (T0) and the 6-month follow-up (T6). A two-sample t-test was used to compare the delta values across the classifications. The following *p*-values were observed for females: ΔFG, *p* = 0.33; ΔCr, *p* = 0.28; ΔTC, *p* = 0.04; ΔHDL-C, *p* = 0.35; ΔTG, *p* = 0.25; ΔLDL-C, *p* = 0.05; ΔAST, *p* = 0.06; ΔALT, *p* = 0.20. Significant differences were found for ΔTC and ΔLDL-C in the female sample.

The gender comparison for the effectiveness of different sports shows significant gender differences in metabolic responses, particularly in ΔTC and ΔLDL-C, with Z-scores above 1.5 in some groups. ΔCr and ΔTG also showed significant variations, indicating gender-specific responses in creatinine and triglyceride levels after exercise. Males experienced more substantial improvements in cholesterol and creatinine levels with anaerobic sports, whereas females benefited more from aerobic activities, particularly with regard to triglycerides and markers of liver function such as ΔAST and ΔALT.

Figure 5 represents the significant normalised gender differences (Z-scores > 1) in metabolic delta values across various sports classifications. Z-scores reflect how many standard deviations a value is from the mean, allowing comparison of relative differences across variables. A Z-score above 1 indicates a difference that exceeds one standard deviation between male and female participants. The *p*-values for the differences across the variables are as follows: ΔFG, *p* = 0.03; ΔCr, *p* = 0.02; ΔTC, *p* = 0.01; ΔHDL-C, *p* = 0.04; ΔTG, *p* = 0.03; ΔLDL-C, *p* = 0.01; ΔAST, *p* = 0.05; ΔALT, *p* = 0.02. The statistical test used was a two-sample *t*-test for comparing the means between males and females.

## 4. Discussion

Our study is distinguished by its specific approach in analysing gender differences in metabolic response to a combined MD and physical activity intervention, focusing on how these factors distinctly influence glycaemic, lipid and liver function parameters between men and women. A noteworthy finding is that the observed weight loss exceeded the predictions made by both POUNDS Lost and CALERIE™ Phase 2 models. This discrepancy highlights the potential variability in adherence, individual metabolic differences and the inherent limitations of the predictive models, which may not fully account for the complexity and diversity of the study population [11].

While previous studies have generally examined the overall effects of the MD or physical activity [16,17,18], few have stratified the analysis by gender to assess the specific impact on metabolic parameters [19,20,21]. In addition, our study provides a detailed analysis of how different types of physical activity (aerobic, anaerobic and combined) distinctly modulate lipid and glycaemic profiles.

This study shows significant gender differences in metabolic responses to the combined MD and exercise intervention, particularly in lipid metabolism and modulation of liver enzymes. The intervention produced greater reductions in TC and LDL-C among men, as well as a marked decrease in ALT [22]. Women, in contrast, showed a significant increase in HDL-C levels, a protective factor for cardiovascular health, which appears to be influenced by oestrogen and its role in increasing HDL [23]. These results are in line with previous studies reporting gender-specific metabolic responses due to hormonal and physiological differences [24].

The six-month intervention based on the MD combined with physical activity led to significant reductions in fasting glucose levels in both men and women, demonstrating the effectiveness of the programme in improving glycaemic control. This evidence is in line with studies by Ahmad et al. [25], who showed a 30% reduction in the risk of type 2 diabetes with adherence to the MD, mainly due to an improvement in insulin resistance and a reduction in inflammatory markers. Bédard et al. [26] also found favourable effects on glycaemic control, noting that in men the increased insulin sensitivity was probably influenced by superior muscle mass, which facilitates glucose absorption and optimises the metabolic response. The improvement in fasting blood glucose observed in both sexes underlines the metabolic flexibility induced by the MD and physical activity.

Lipid profiles improved significantly in both sexes, with reductions in TC and LDL-C being more pronounced in men, consistent with hormonal influences and metabolic adaptations to the intervention. This could be attributable to hormonal influences, as higher testosterone levels in men promote faster lipid metabolism and more effective mobilisation of lipid reserves during dietary changes [27]. In contrast, a significant increase in HDL levels was observed in women, which is relevant for cardiovascular protection and probably influenced by oestrogen, which plays a key role in increasing HDL [28]. These results are in line with data from the PREDIMED-Plus trial, in which similar effects were observed, suggesting that gender-differentiated dietary approaches could optimise the cardiovascular benefits associated with the MD [29]. Furthermore, Adeniyi et al. [30] showed that a physical activity programme combined with dietary interventions further improved lipid profiles: in men, greater reductions in total cholesterol and LDL-C were observed, whereas in women, significant increases in HDL supported cardiovascular health benefits.

Both aerobic and anaerobic activities significantly influenced TC and LDL levels, reducing cardiovascular risk in both sexes, with gender-specific adaptations reflecting hormonal and physiological differences [31]. Interestingly, the reduction in fasting glucose was consistent regardless of the type of physical activity. The combination of dietary adherence and regular physical activity improves glucose homeostasis by improving insulin sensitivity, reducing systemic inflammation and optimising energy utilisation. However, the gender-stratified analysis showed that men benefited more from anaerobic activities, as also suggested by research linking resistance training to greater improvements in lipid metabolism in men, probably influenced by testosterone [32]. Conversely, women exhibited a significant increase in HDL in response to aerobic activity, a result that reflects evidence attributing a key role in HDL modulation to this type of exercise, especially in the presence of female hormonal influences [33]. The significant reduction in fasting blood glucose, regardless of the type of physical activity, is consistent with other research confirming the positive effect of physical activity and the Mediterranean diet on insulin sensitivity. Studies such as that of Ahmad et al. [34] demonstrated a 30% reduction in diabetes risk with adherence to the MD, with glycaemic improvements generally independent of the type of exercise. In terms of lipid metabolism, our work confirms the reductions in TC and LDL-C with both types of activity, with a more pronounced response of men to anaerobic exercise, an effect already observed and attributed to the influence of testosterone on lipid metabolism [29]. Conversely, the significant increase in HDL in women with aerobic activity is corroborated by evidence showing a particular efficacy of aerobic exercise in increasing HDL in the female context [35]. The reductions in AST and ALT observed with exercise, and particularly the more pronounced decrease in ALT in men, are consistent with studies showing a positive hepatic response to exercise, especially in men with higher visceral fat stores, who respond better to anaerobic stimuli [36]. Overall, these results underline the importance of considering gender-specific responses when designing interventions combining the MD and physical activity, as they may optimise metabolic outcomes for both men and women.

Although this study produced important results, several limitations must be considered. The use of unvalidated questionnaires may limit the generalisability of the results, and the self-selection process of the physical activity groups could introduce a selection bias, as participants chose groups based on their preferences and habits, which could limit the applicability of the results to populations with different activity levels or motivations. Furthermore, recruiting participants from an obesity clinic, while advantageous in terms of ensuring motivated and informed individuals, may result in a different sample from the general population in terms of health awareness and baseline behaviours. Furthermore, while BIA was used for body composition assessment, the Tanita BC-420 MA device is based on proprietary algorithms, and it is unclear whether the population used to develop these algorithms corresponds to the population of this intervention study. Furthermore, DXA, which is used as a reference method for validation in some contexts, is considered a reference method rather than a gold standard, and therefore, BIA results cannot be validated directly against DXA [15]. 

## 5. Conclusions

The intervention produced significant metabolic benefits in both men and women, but gender-specific responses indicate that the efficacy of interventions could be maximised through individualised strategies. These findings, as summarised in Table 3 (take-home messages), support the growing evidence for integrating individualised nutritional and exercise strategies and suggest that health professionals should consider gender as a critical factor in optimising therapeutic outcomes.

## Figures and Tables

**Figure 1 nutrients-17-00354-f001:**
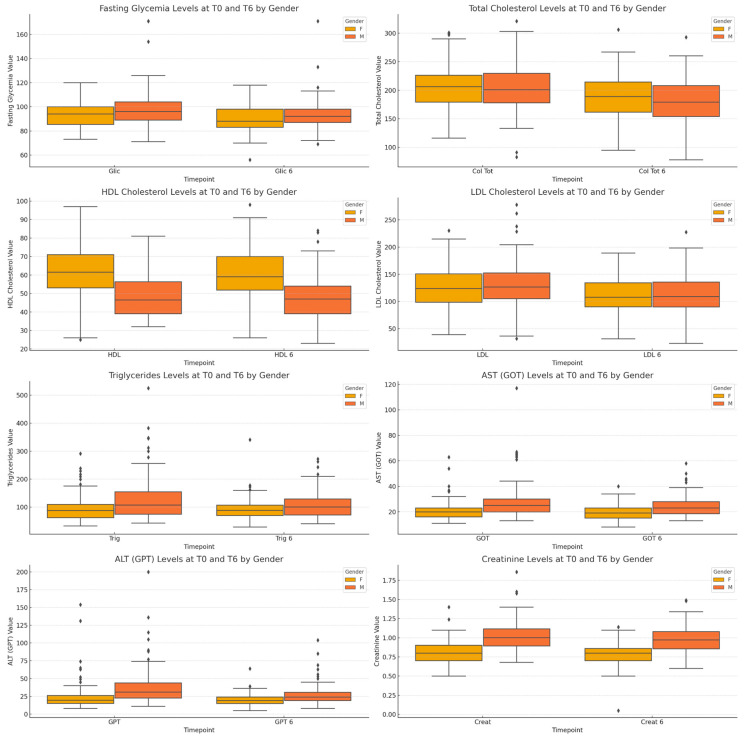
Comparison of fasting glycaemia, total cholesterol, HDL, LDL, triglycerides, AST (GOT), ALT (GPT) and creatinine levels at T0 and T6 by gender.

**Figure 2 nutrients-17-00354-f002:**
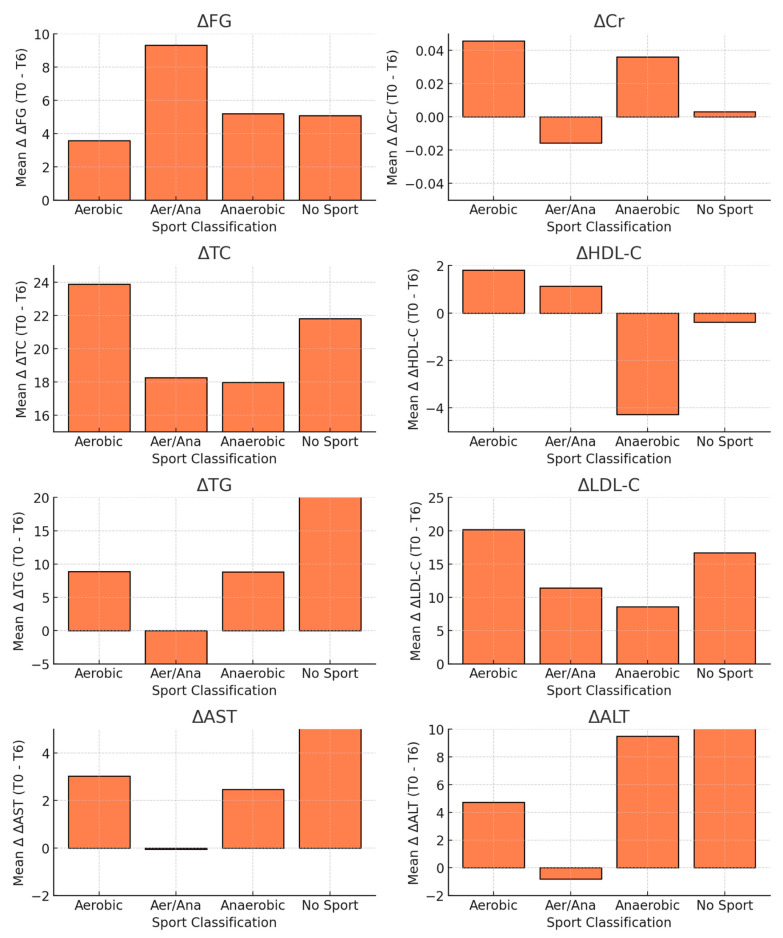
Mean Δ (T0–T6) of metabolic parameters across sport classifications.

**Figure 3 nutrients-17-00354-f003:**
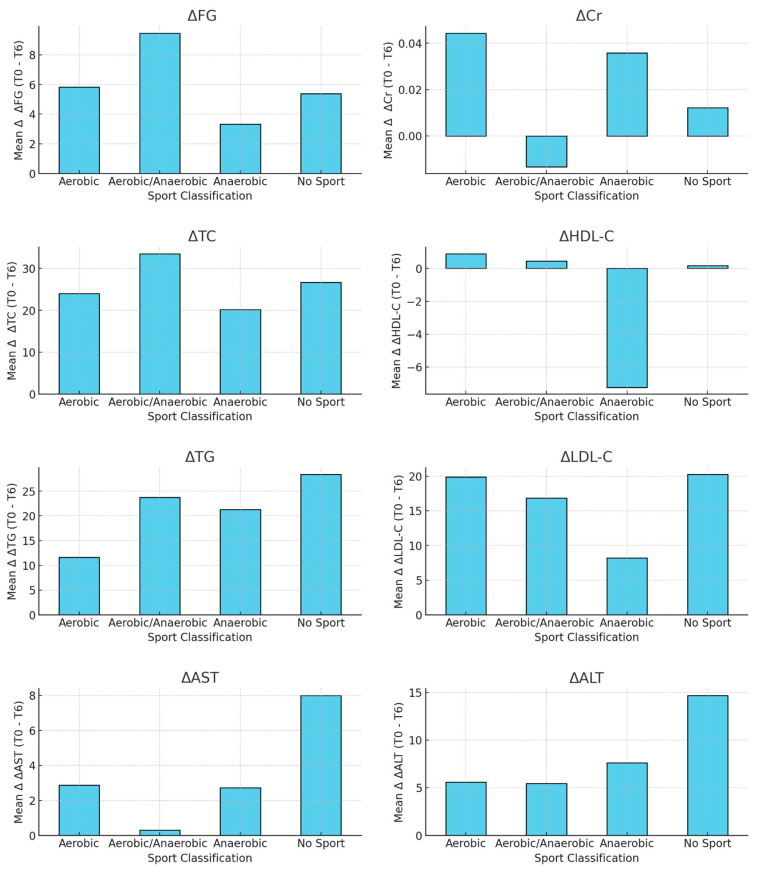
Mean Δ (T0–T6) of metabolic parameters across sport classifications in males.

**Figure 4 nutrients-17-00354-f004:**
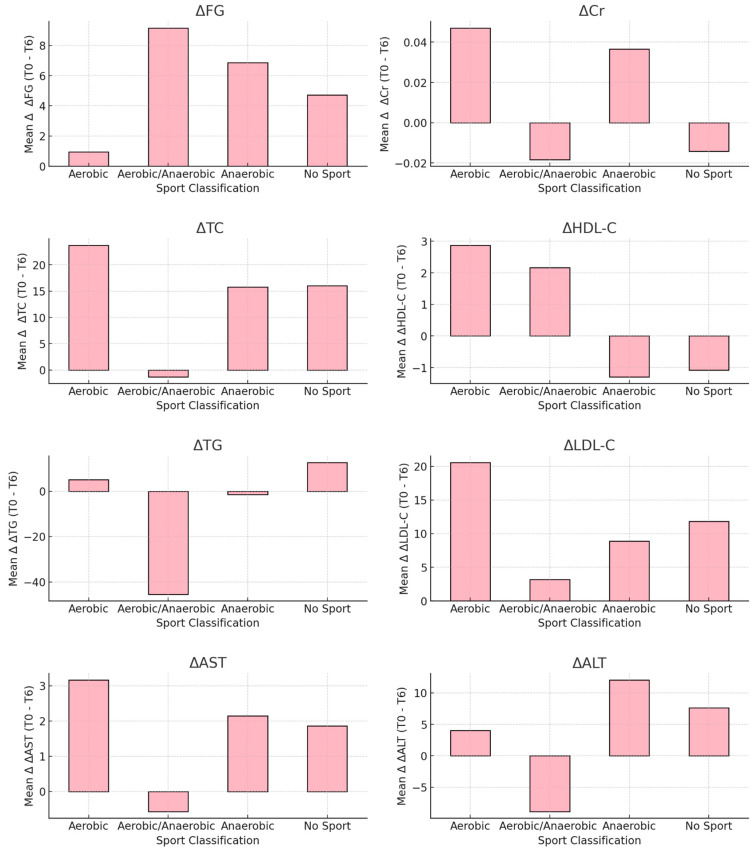
Mean Δ (T0–T6) of metabolic parameters across sport classifications in females.

**Figure 5 nutrients-17-00354-f005:**
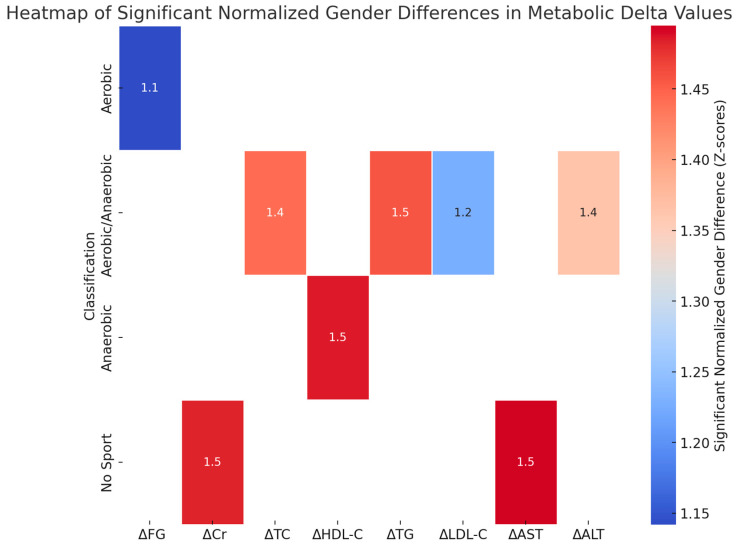
Heatmap of significant normalised gender differences in metabolic delta values across sports classifications.

**Table 1 nutrients-17-00354-t001:** Baseline characteristics of the study population by gender (*n* = 205).

Variable	Total (*n* = 205)	Male (*n* = 107)	Female (*n* = 98)	*p*-Value
Age (years)	48.4 ± 12.9	48.6 ± 13.5	48.2 ± 12.3	0.8265
BMI (kg/m^2^)	29.8 ± 5.2	30.6 ± 5.5	28.8 ± 4.8	0.0128
FM (kg)	27.9 ± 11.2	27.0 ± 12.4	29.0 ± 9.8	0.2076
FFM (kg)	54.3 ± 11.3	63.2 ± 7.7	44.6 ± 4.8	<0.0001
FMI (kg/m^2^)	9.8 ± 3.8	8.8 ± 3.9	11.0 ± 3.5	0.0001
FFMI (kg/m^2^)	18.8 ± 2.8	20.4 ± 2.7	16.9 ± 1.6	<0.0001
AC (cm)	102.5 ± 13.0	106.3 ± 13.1	98.5 ± 11.7	<0.0001
BMR (kcal/day)	1712.1 ± 345.6	1965.3 ± 268.4	1437.8 ± 160.9	<0.0001
Smoker (%)	17%	12%	23%	0.033
Income (%)				
<EUR 20,000	29%	32%	26%	0.541
EUR 20,000–EUR 40,000	44%	41%	47%	0.614
EUR 40,000–EUR 60,000	18%	19%	17%	0.726
>EUR 60,000	9%	8%	10%	0.833
Category Work (%)				
Sales and Services	49%	51%	47%	0.489
Professional Services	20%	19%	21%	0.741
Healthcare and Wellness	15%	14%	16%	0.615
Other	16%	16%	16%	0.922
Sport				
Do you play a sport? (%)				0.5272
Yes	50%	54.9%	45.1%	
No	50%	49.5%	50.5%	
Sport hours per week (%)				0.0088
<5 h	72%	45.8%	54.2%	
5–10 h	25%	73.1%	26.9%	
>10 h	3%	100%	0%	

Table 1 presents the baseline characteristics of the study population (*n* = 205), stratified by gender (107 males and 98 females). Continuous variables are expressed as means and standard deviations. Categorical variables are presented as percentages. BMI—body mass index (kg/m^2^); FM—fat mass (kg); AC—abdominal circumference (cm); FFM—fat-free mass (kg); FMI—fat mass index (kg/m^2^); FFMI—fat-free mass index (kg/m^2^); BMR—basal metabolic rate (kcal/day). For continuous variables, a t-test was used to compare gender differences, and for categorical variables, a chi-square test was applied. Statistical significance was set at *p* ≤ 0.05.

**Table 2 nutrients-17-00354-t002:** Descriptive statistics, deltas and *p*-values for metabolic parameters at baseline and after 6 months (T6) stratified by gender.

Balanced Group	TOTAL		MALE		FEMALE			TOTAL Δ (T0–T6)		MALE Δ (T0–T6)		FEMALE Δ (T0–T6)		
n.	Mean	std	Mean	SD	Mean	SD	m vs. f *p*-values	Mean	std	Mean	SD	Mean	SD	*p* Δ males vs. females
BMI T0	29.8	5.2	30.6	5.5	28.8	4.8	0.0128							
BMI T3	28.0	5.0	28.8	5.2	27.0	4.5	0.0081							
BMI T6	26.9	5.1	27.9	5.0	25.8	4.9	0.0022	−2.8	1.9	−2.7	1.4	−3	2.3	0.201
FM T0	27.9	11.2	27.0	12.4	29.0	9.8	0.2076							
FM T3	24.1	10.7	23.0	11.8	25.3	9.3	0.1342							
FM T6	22.1	10.0	21.0	10.8	23.4	9.0	0.0853	−5.8	3.8	−6	4.1	−5.6	3.3	0.4059
AC T0	102.5	13.0	106.3	13.1	98.5	11.7	<0.0001							
AC T3	96.8	12.4	100.7	13.1	92.4	10.1	<0.0001							
AC T6	93.9	11.8	97.7	12.1	89.9	10.0	<0.0001	−8.1	4.7	−8.2	4.4	−8	5.1	0.73
FFM T0	54.3	11.3	63.2	7.7	44.6	4.8	<0.0001							
FFM T3	53.1	10.9	61.7	7.5	43.8	4.4	<0.0001							
FFM T6	52.5	11.1	61.3	7.5	42.9	4.3	<0.0001	−1.8	2.1	−1.9	2.4	−1.7	1.7	0.442
Body Water (kg)	40.4	8.7	47.1	6.0	33.2	4.0	<0.0001							
Body Water T3	39.3	8.3	45.7	5.8	32.3	3.7	<0.0001							
Body Water T6	38.7	8.2	45.1	5.7	31.9	3.8	<0.0001	−1.6	2.1	−2	1.9	−1.3	2.2	0.023
FMI T0	9.8	3.8	8.8	3.9	11	3.5	0.0001							
FMI T6	7.6	3.5	6.8	3.4	8.8	3.3	<0.0001	−2	1.3	−2	1.3	−2.1	1.3	0.4184
FFMI T0	18.8	2.8	20.4	2.7	16.9	1.6	<0.0001							
FFMI T6	18.1	2.7	19.8	2.6	16.3	1.4	<0.0001	−0.6	0.7	−0.6	0.8	−0.6	0.6	0.8705
BMR T0	1712.1	345.6	1965.3	268.4	1437.8	160.9	<0.0001							
BMR T3	1654.5	347.2	1904.5	252.6	1377.8	193.4	<0.0001							
BMR T6	1632.7	332.1	1879.9	255.2	1362.1	139.0	<0.0001	−78.4	62.9	−85.4	70.2	−70.1	53.1	0.098

T0 (baseline), T3 (3-month follow-up) and T6 (6-month follow-up) measurements of body composition metrics, including body mass index (BMI, kg/m^2^), fat mass (FM, kg), abdominal circumference (AC, cm), fat-free mass (FFM, kg), fat mass index (FMI, kg/m^2^), fat-free mass index (FFMI, kg/m^2^), body water (%) and basal metabolic rate (BMR, kcal/day). Δ indicates changes from T0 to T6. Mean and standard deviation (SD) values are shown for the overall group and separately for males and females. Statistical significance (*p*-values) for differences between genders at T0, T3, T6 and Δ are included, highlighting notable gender differences in FMI and FFMI.

**Table 3 nutrients-17-00354-t003:** Gender-specific metabolic responses and clinical take-home messages.

Topic	Our Findings	Take-Home Message
Gender Differences	Men showed more significant reductions in TC, LDL-C and ALT, while women had a notable increase in HDL levels.	Men benefit from lipid and liver function improvements, while women experience enhanced HDL levels, contributing to cardiovascular protection.
Effects of Physical Activity	Anaerobic activities led to greater reductions in TC and LDL in men, whereas aerobic activity significantly increased HDL in women.	Tailored activity types (anaerobic for men, aerobic for women) may maximise lipid profile benefits.
Glycaemic Control	Both genders exhibited reductions in fasting glucose with the intervention, regardless of activity type.	The MD combined with physical activity effectively improves fasting glucose control across genders.
Liver Function	Reductions in AST and ALT were observed, with men showing a more pronounced decrease in ALT.	Physical activity improves liver enzyme levels, with men showing a stronger hepatic response.
Renal Function	No significant changes in creatinine levels across activity types.	Physical activity combined with the MD is safe for renal function within the intervention’s timeframe.
Clinical Implications	Our gender-stratified analysis highlights specific metabolic responses to different activity types.	Personalised, gender-specific intervention strategies can enhance metabolic and cardiovascular outcomes.

## Data Availability

The data supporting the findings of this study are available from the corresponding author upon reasonable request. All data will be shared in a de-identified format to protect participant confidentiality.

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
