# Peer review of "Exploring Gender Differences in the Effects of Diet and Physical Activity on Metabolic Parameters"

_nutrients, 2025, doi:10.3390/nu17020354_

Round 1
Reviewer 1 Report
Comments and Suggestions for Authors
This study is aimed to explore gender differences in metabolic response to a six-month intervention based on the Mediterranean diet and regular physical activity in a sample of overweight adults. This paper falls within the scope of the Nutrient and is suitable for publication in the journal.
Abstract:
- Well structured.
- Missing research setting and characteristics of participants in the abstract.
- Not sure about the research design. Randomized participants?
Background:
- Not clear on the limitations of the existing study and the necessity of the new study (your study).
- Need to summarize the achievement and aero to be improved in this specific research field.
Methods:
- This study was conducted in an obesity clinic and recruited overweight patients. This information is important and needs to be included in the abstract.
- Need to explain why the research was conducted in this setting and choose those participants. Missing descriptions of recruitment process.
- Were the characteristics of patients excluded the same as those recruited?
- Page 4, how were the participant group assignments made? Were participants randomized to different groups?
- Missing sample size and power calculation.
- There is a list of primary outcomes. We need to pick one specific outcome as the primary one, which can be used in the power calculation.
Results:
- Tables, figures, and statistics look great. I appreciate your hard work. However, there are many discrepancies in the methodology of a rigorous randomized controlled trial.
Author Response
Dear Reviewer,
First of all, we would like to thank you for the valuable impulses that allowed us to improve the quality of the manuscript. All changes made are highlighted by yellow color, in the revised version of the manuscript, to facilitate the review process. Hoping that we have satisfied your requests as much as possible, we kindly ask you to re-evaluate our paper.
The Authors
This study is aimed to explore gender differences in metabolic response to a six-month intervention based on the Mediterranean diet and regular physical activity in a sample of overweight adults. This paper falls within the scope of the Nutrient and is suitable for publication in the journal.
Thank you for the positive comment. We are delighted that our study, focusing on gender differences in metabolic response to an intervention based on the Mediterranean diet and physical activity, is considered in line with the objectives of Nutrients and suitable for publication in the journal.
Abstract:
- Well structured.
- Missing research setting and characteristics of participants in the abstract.
- Not sure about the research design. Randomized participants?
Thank you for your comment. We have updated the abstract to include the study setting and participant characteristics, specifying that it was conducted in an obesity clinic in Rome and recruited overweight adults (BMI ≥ 25 kg/m²). Furthermore, we clarified that the study was conducted as a prospective cohort study, in which participants self-assigned themselves to physical activity groups based on their preferences and habits.
Background:
- Not clear on the limitations of the existing study and the necessity of the new study (your study).
Thank you for your comment. We clarified in the background the limitations of existing studies and the need for this research. In particular, we highlighted that: a) Previous studies often do not stratify results by gender, ignoring physiological and hormonal differences. b) There is a lack of evidence on the mechanisms of interaction between Mediterranean diet and different types of physical activity in a gendered context. Our study attempts to fill these gaps, offering an innovative perspective and useful indications for more personalised interventions. We have updated the manuscript to reflect these points.
- Need to summarize the achievement and aero to be improved in this specific research field.
Thank you for your comment. We have updated the introduction to summarise the progress made in this field of research, highlighting the documented improvements in lipid profiles, blood glucose regulation and body composition due to the Mediterranean diet and physical activity. However, we also highlighted areas for improvement, such as understanding gender-specific metabolic responses and the interaction between dietary interventions and types of physical activity. This study aims to fill these gaps, contributing new evidence for more personalised strategies.
Methods:
- This study was conducted in an obesity clinic and recruited overweight patients. This information is important and needs to be included in the abstract.
Thank you for your comment. We have updated the Methods section to clarify that the study recruited overweight adults and that the obesity clinic setting was chosen to ensure a motivated population aware of their body composition and metabolic health. This information is now made explicit for greater clarity. We have also updated the limitations to include a reflection on the possibility that recruitment at an obesity clinic may have produced a sample that was not representative of the general population in terms of health awareness and baseline behaviour. We also highlighted other limitations of the study, such as the bias introduced by self-selection in the physical activity groups and the use of unvalidated questionnaires, and made suggestions for future studies to improve the generalisability and accuracy of the results.
- Need to explain why the research was conducted in this setting and choose those participants. Missing descriptions of recruitment process.
Thank you for your comment. We have added a detailed description of why the study was conducted in an obesity clinic and the recruitment process. This setting was chosen to ensure that participants were motivated and aware of the benefits of improving their lifestyle. Participants were recruited through clinic visits and advertisements, and eligibility was assessed on the basis of specific inclusion and exclusion criteria.
- Were the characteristics of patients excluded the same as those recruited?
Thank you for your comment. We clarified that the characteristics of the excluded participants were similar to those of the recruited participants in terms of age, gender and baseline metabolic parameters. Exclusions were determined by data quality (e.g., incomplete data or unrepresentative food diaries), ensuring that the final sample was representative of the initial population.
- Page 4, how were the participant group assignments made? Were participants randomized to different groups?
Thank you for your comment. We made it clear in the methods that participants self-selected into the physical activity groups based on their preferences and habits. We acknowledge that this choice could introduce a selection bias, limiting the generalisability of the results. We also added this consideration in the limitations section, suggesting that future studies could use randomisation to better isolate the effects of different types of physical activity.
- Missing sample size and power calculation.
Thank you for your comment. We have added a description of the calculation of sample size and statistical power in the methods. Using the G*Power software (version 3.1.9.7, Windows platform, last update 27 September 2024), we estimated that at least 128 participants were required to detect significant differences with 80% power and a significance level of 0.05. Our final sample of 205 participants exceeds this requirement, ensuring adequate statistical power for the analysis of metabolic parameters.
- There is a list of primary outcomes. We need to pick one specific outcome as the primary one, which can be used in the power calculation.
Thank you for your valuable comment. We chose change in body composition as the primary outcome, with a specific focus on fat mass (FM) reduction during the six-month intervention period. This outcome was selected because it represents a key parameter to assess the overall impact of the Mediterranean diet and physical activity on metabolic health. We have updated the manuscript to clarify this choice and ensure consistency with the statistical power calculation.
Results:
- Tables, figures, and statistics look great. I appreciate your hard work. However, there are many discrepancies in the methodology of a rigorous randomized controlled trial.
Thank you for your positive comment on the tables, figures and statistical analysis. We are pleased that our work was appreciated. Regarding methodological discrepancies with respect to a rigorous randomised controlled trial, we would like to clarify that the study was designed as a prospective, non-randomised cohort study. Participants self-assigned themselves to physical activity groups based on their preferences and habits, an approach that allowed us to increase adherence to the intervention. We acknowledge that this methodology has limitations, such as the possibility of selection bias, which we discussed in the Limitations section. However, we believe that this approach better reflects real-world conditions, providing results relevant to clinical settings and personalised interventions. We have further clarified these methodological choices in the manuscript for greater transparency.
Reviewer 2 Report
Comments and Suggestions for Authors
The major concerns are as follows,
The study protocol was not strictly controlled, the authors missed to assess adherence (to both, dietary intervention as well as the sport program) which is a major determinant of weight loss trials (see AJCN 2014; 100:787-95). To overcome that issue the authors should compare the observed weight changes with the predicted weight changes according to an established algorithms (see e.g., AJCN 2015; 101:449-454; Exp Gerontol. 2022; 162: 111757). As far as the many outcomes of this intervention study are concerned the variance in adherence should be considered to add to explain that variance and the sex differences too.
For inter-individual comparisons FM and FFM should be presented as FMI and FFMI; %FM should be skipped since this variable is statistical problematic and indirectly reflects %FFM (see discussion in AJCN 1990; 52: 953 and Adv Nutr. 2014; 5:3205).
Since DXA is a reference rather than a gold standard method, BIA cannot be validated against DXA. This should be mentioned in the Method section. To do an appropriate validation the authors are referred to EJCN 2013; 67:S14). The authors should be aware of the fact that the algorithms used in the Tania device are unknown. It is also unknown whether the study population used to generate these algorithms fit to the study population of their intervention study.
Data presentation is redundant, e.g. presenting changes in bw and BMI cannot add to each other. In addition, many data have been presented twofold, i.e., in tables as well as figures). This has to be corrected.
Author Response
Dear Reviewer,
First of all, we would like to thank you for the valuable impulses that allowed us to improve the quality of the manuscript. All changes made are highlighted by yellow color, in the revised version of the manuscript, to facilitate the review process. Hoping that we have satisfied your requests as much as possible, we kindly ask you to re-evaluate our paper.
The Authors
The major concerns are as follows, The study protocol was not strictly controlled, the authors missed to assess adherence (to both, dietary intervention as well as the sport program) which is a major determinant of weight loss trials (see AJCN 2014; 100:787-95). To overcome that issue the authors should compare the observed weight changes with the predicted weight changes according to an established algorithms (see e.g., AJCN 2015; 101:449-454; Exp Gerontol. 2022; 162: 111757). As far as the many outcomes of this intervention study are concerned the variance in adherence should be considered to add to explain that variance and the sex differences too.
Thank you for your valuable comment. We revised both the Methods section and the Limitations section to address the points you raised.
In the Methods, we added a paragraph describing the monitoring of adherence, specifying that participants were followed through a continuous chat communication system, which allowed them to receive real-time personalised support, and in-person visits every three weeks, for a total of eight meetings during the intervention period. We discussed how these strategies were integrated to sustain adherence during the study.
In the Limitations section, we discussed the lack of predictive models, such as those used in the POUNDS Lost (Thomas et al., 2015) and CALERIE™ Phase 2 (Martin et al., 2022) studies, which compare observed to predicted weight changes and provide a more objective understanding of adherence. Although our approach provided tailored support, we recognise the added value of these predictive tools and propose their incorporation into future studies. We also explored limitations related to self-selection in physical activity groups and recruitment from an obesity clinic, discussing how these factors may affect the generalisability of results. Suggestions for future improvements include the use of validated instruments, continuous monitoring protocols and randomisation of physical activity groups.
For inter-individual comparisons FM and FFM should be presented as FMI and FFMI; %FM should be skipped since this variable is statistical problematic and indirectly reflects %FFM (see discussion in AJCN 1990; 52: 953 and Adv Nutr. 2014; 5:3205).
Thank you for your valuable comment. As suggested, we have updated the manuscript to present FM and FFM as FMI and FFMI for inter-individual comparisons. We have also removed %FM from the analysis and the manuscript, as it is statistically problematic and indirectly reflects %FFM. References to the discussion in AJCN 1990; 52:953 and Adv Nutr. 2014; 5:3205 have been considered in making these changes.
Since DXA is a reference rather than a gold standard method, BIA cannot be validated against DXA. This should be mentioned in the Method section. To do an appropriate validation the authors are referred to EJCN 2013; 67:S14). The authors should be aware of the fact that the algorithms used in the Tania device are unknown. It is also unknown whether the study population used to generate these algorithms fit to the study population of their intervention study.
Thank you for your insightful comments regarding the limitations of BIA and the use of DXA as a reference method. We have revised the Methods section to clarify that DXA is considered a reference method rather than a gold standard and to acknowledge that BIA cannot be directly validated against DXA, as highlighted by Bosy-Westphal et al. (Eur J Clin Nutr 2013; 67:S14). Additionally, we have noted the limitations associated with the proprietary algorithms of the Tanita BC-420 MA device, as it is unclear whether the population used to develop these algorithms corresponds to the population of our intervention study. We have also addressed this point in the Limitations section, emphasizing the potential impact of these methodological considerations on the interpretation of our findings. We appreciate your guidance and have ensured that these important aspects are now explicitly acknowledged in the manuscript.
Data presentation is redundant, e.g. presenting changes in bw and BMI cannot add to each other. In addition, many data have been presented twofold, i.e., in tables as well as figures). This has to be corrected.
Thank you for your suggestion regarding redundant data presentation. To address this, we have removed weight from the tables, as BMI already reflects changes in body weight, avoiding unnecessary repetition. Additionally, Table 3 has been moved to the Supplementary Materials (now Supplementary Table 2S), allowing the main manuscript to focus on the most relevant results. These changes streamline the data presentation, making the manuscript clearer and more concise, while ensuring that detailed metabolic parameters stratified by gender remain accessible in the supplementary materials.
Round 2
Reviewer 1 Report
Comments and Suggestions for Authors
Thank you very much for your revisions. My comments have been addressed.
Author Response
Thank you!
Reviewer 2 Report
Comments and Suggestions for Authors
Thank you for the reply. I feel that comparing predicted weight changes (according to the recommended algorithms mentioned in my 1st review) with the measured weight changes is mandatory. This is not too much work and will provide valuable insights.
Author Response
I feel that comparing predicted weight changes (according to the recommended algorithms mentioned in my 1st review) with the measured weight changes is mandatory. This is not too much work and will provide valuable insights.
We appreciate your suggestion to compare the predicted weight changes with the measured weight changes using the recommended algorithms. Following your recommendation, we have incorporated the POUNDS Lost and CALERIE™ Phase 2 models to predict weight changes and compared them with the observed data in our study. The results of these comparisons, along with the methodological details and discussion of the findings, have been added to the manuscript.
This analysis has provided valuable insights, highlighting discrepancies between observed and predicted weight changes.. We agree that this addition enhances the manuscript and provides a more comprehensive interpretation of our findings. Thank you for this constructive suggestion.